# Contrasting Public and Scientific Assessments of Fracking

**Yu Zhang \*, John A. Rupp and John D. Graham**

O'Neill School of Public and Environmental Affair, Indiana University Bloomington,
Bloomington, IN 47405, USA; rupp@indiana.edu (J.A.R.); grahamjd@indiana.edu (J.D.G.)

\* Correspondence: zhangyu236@gmail.com

**Abstract:** This paper examines whether public perceptions of the claimed advantages and disadvantages of fracking are consistent with an evidence-based assessment of the claimed advantages and disadvantages. Public assessments are obtained from an internet-based opinion survey in 2014 in six states: California, Illinois, New York, Ohio, Pennsylvania, and Texas. The survey presented eleven advantages and eleven disadvantages of fracking derived from local media stories, from advocacy claims made by pro- or anti-fracking groups, and from think tank pieces. Then the respondents were asked to indicate their feelings about how important each claimed advantage and disadvantage was to their support of/opposition to fracking. Scientific assessments regarding the same claims are compiled from available peer-reviewed literature and evidence-based reviews. We classify each claim as either (a) supported by the weight of the available evidence, (b) not supported by the weight of the available evidence, or (c) there is inadequate evidence to assess it. We find less consistency with respect to the disadvantages than advantages. Respondents perceive four disadvantages out of eleven as extremely important while there is inadequate evidence to assess them or the available evidence does not support them. Our comparison has interesting implications for understanding the controversy about fracking.

**Keywords:** fracking; advantages and disadvantages; public perceptions; scientific assessment

## 1. Introduction

Horizontal drilling and hydraulic fracturing are spawning a new era of natural gas production. Hydraulic fracturing, also referred to as "fracking" is a well-stimulation technique that injects water, sand, and chemicals into a bedrock formation via the well under high pressure to create fractures in the rock. It has been used widely in tight sandstone and shale to increase not only gas but also oil production [1]. The rapid pace of extraction of previously inaccessible shale gas resources is triggering spirited public debates about the advantages and disadvantages of expanded gas production. Public opinion is playing an increasingly important role in the future of fracking. Ballot propositions in local U.S. communities have reached conflicting results (e.g., a ban in Denton, Texas, but permission to proceed in Santa Barbara, California).

Several dozen public perception surveys of fracking have been conducted between 2010 and 2015 [2–4]. The surveys establish several patterns: public awareness of fracking varies across states but is often quite low; incorrect knowledge of unconventional methods is common; and there is an increasing public opposition to fracking [2]; and most respondents support greater regulation of fracking [4,5].

A wealth of public opinion studies of controversial technologies has studied the factors shaping public perceptions. We summarized the main factors studied in social science research as follows: The psychometric risk-perception paradigm developed in the 1970s is the first body of literature explaining public risk perceptions. The paradigm posits that risk perceptions of lay people are based less on reason and analysis and more on two broad characteristics of risk: dread and the unknown. "Dread risk" relates to factors such

as the lack of individual control of a risk, the potential for catastrophic consequences, and the lack of familiarity of hazards and perceptions of inequity in the distribution of risks and benefits. "Unknown risk" refers to effects that are unobservable, unknown, new, and potentially delayed in their manifestation of harm [6]. The risk-perception framework has been replicated on independent data [7], applied in numerous case studies [8], and is a centerpiece of textbooks on risk perceptions [9,10]. However, recent research found that the explanatory power of psychometric factors is smaller when analyzing individual-level data than when exploring differences in average risk ratings for groups of individuals for a set of different hazards [11]. Douglas et al. [12] and Kahan [13] believed that worldview might explain interpersonal differences of perceptions of risks better. Four types of worldviews have been constructed by various measures: hierachist, individualist, egalitarian, and fatalist. Boudet et al. [3] found that egalitarian worldviews are negatively associated with support for fracking. Public opinion scholars have sought to analyze factors shaping public perceptions of emerging shale gas expansion more broadly. An array of sociodemographic literatures identified that women and minority groups are more likely to have a higher perception of risks and are less supportive of emerging technologies [3,14–16]. Previous studies found mixed effects of age, income, and education on perceptions of shale gas development [3,17,18]. Geographic proximity or location is another factor getting attention, yet scholars found a mixed effect [2,19,20]. Royalty payments or community reinvestments might attenuate public perceptions of risks. For example, Paydar et al. [21] found that community reinvestments may switch part of opponents of fracking to supporters. Last, trigger events of a new technology then amplified by social media might shape public perceptions [22]. Though no empirical evidence shows the effects of trigger events of fracking, the media reports about drinking water contamination due to methane and earthquakes related to waste water injection are causing public concerns and even regulatory changes [23].

This research attempted to understand public perceptions of fracking and public controversy of fracking. This paper sheds light on whether there is need for public education about science and economics of fracking. We seek to determine whether public perceptions of the claimed advantages and disadvantages of fracking are consistent with an evidence-based assessment of the claimed advantages and disadvantages. Please note that an evidence-based assessment is different from a survey of expert perceptions because a variety of non-evidence factors may influence an expert's perception of the advantages and disadvantages of a technology. Experts are human beings, and there is a wealth of studies suggesting that values as well as evidence play a role in expert perceptions of technology. Our comparison of public perceptions with evidence has interesting implications for understanding the controversy about fracking. If public perceptions about fracking are consistent with the evidence, then the fracking controversy is unlikely to be diminished by bringing more evidence to the public debate. If some of the perceptions are not consistent with the evidence, then it is possible that bringing more evidence to the public debate could reduce controversy.

We started by reporting the results of an online survey in six states where fracking is technically feasible or underway. In the survey, we offered a neutral narrative and visual definition of fracking, thereby explaining what the terms "hydraulic fracturing" and "fracking" mean. We then asked for respondents' perceptions of a series of advantages and disadvantages of fracking. Separate from the survey, we compiled, evaluated, and classified the weight of scientific evidence regarding each claimed advantage or disadvantage. Last, we compared the public and scientific assessments of each claim for possible disconnections. We attempted to explain why respondents might regard some claimed disadvantages as highly important, even though the weight of the scientific evidence is inadequate to assess them or does not support them.

## 2. Materials and Methods

### 2.1. A Survey on Public Perceptions of the Advantages and Disadvantages of Fracking

To assess public perceptions, we considered random digit dial, home interview, and online survey methods. Selection criteria were cost, timeliness, response rate, quality of responses, and representativeness of respondents. An online survey, when undertaken properly, is equivalent or superior to the other methods on each of the five criteria [24,25]. The online survey also facilitates randomization of the ordering of questions and the tailoring of questions to the individual respondent, based on their initial responses [26].

We designed the survey instrument and commissioned the GfK Group to conduct the online survey from 3rd June to 2nd July 2014 in six states: California, Illinois, New York, Ohio, Pennsylvania, and Texas. All selected states have geological potential for fracking but various levels of development—from high levels of development in Texas, Pennsylvania, and Ohio to virtually non-existent levels of development in California, Illinois, and New York. The sample of respondents was drawn from GfK's proprietary KnowledgePanel. We obtained 3040 responses from the sample of 5663 residents. After taking missing values and careless answers into consideration, we had a total sample of 2833 for further analysis.

To reduce the effects of any non-response and non-coverage bias in the overall panel, a post-stratification weight was applied based on 2013 demographic data on gender, age, race, education, household income, and metropolitan area from the U.S. Current Population Survey [27]. Certain counties were oversampled where hydraulic fracturing practices ("fracking") are currently used or are considered geologically promising. The weights were also adjusted for this oversampling and the corresponding unequal probabilities of selection in the original design. All the proportions reported in this paper were weighted to ensure statewide representativeness. Table 1 below shows the demographics of the samples from the survey. The unit in each cell is percentage.

**Table 1.** Samples Characteristics.

| Age | Percentage |
|---|---|
| 18–29 | 12 |
| 30–44 | 18.67 |
| 45–59 | 31.91 |
| >60 | 37.42 |
| **Education** | **Percentage** |
| less than high school | 5.75 |
| high school | 24.43 |
| some college | 32.19 |
| bachelor or above | 37.63 |
| **Gender** | **Percentage** |
| male | 47.76 |
| female | 52.24 |
| **Child** | **Percentage** |
| w/children under 18 | 34.43 |
| no children under 18 | 65.57 |
| **Income** | **Percentage** |
| 1st quarter | 26.61 |
| 2nd quarter | 34.06 |
| 3rd quarter | 15.04 |
| 4th quarter | 24.29 |

"Hydraulic fracturing" or "fracking" is derived from the industry term to "frac" a well. While this term within the industry denotes a specific hydraulic process, it is now often used to represent shale gas development as a whole. All respondents were provided with a visual and narrative definition of fracking in the first part of the survey because the term "fracking" is commonly the word of choice in media reports. The next section of the survey outlines respondents' reactions to eleven claimed advantages and eleven claimed disadvantages of fracking. The respondents were asked to indicate their feeling about how important those claimed advantages/disadvantages were to their support of/opposition to fracking (four scales of importance: extremely important, moderately important, slightly important, and not important at all). The survey randomized the order in which the 22 items appeared to the various respondents to minimize the potential bias in the overall results. We derived the claimed advantages and disadvantages from local media stories, from advocacy claims made by pro- or anti-fracking groups, and from think tank pieces. Table 2 presents the wording of the question.

**Table 2.** Survey Question about the Claimed Advantages and Disadvantages of Fracking.

"In the public debate about energy policy, there are advantages and disadvantages that are claimed to be associated with expanding natural gas production in the United States, especially from fracking. For each claim listed below, please indicate how important, if at all, it is in your support of/opposition to fracking: extremely important, moderately important, slightly important, and not important at all."

### Natural Gas Production, Increasingly Accomplished with Fracking

#### Advantages

1) Reduces energy prices, which means cheaper energy for customers

2) Means that the United States can rely less on other countries for energy

3) Creates jobs in exploration and drilling activities

4) Creates jobs in the pipeline and transportation industries

5) Keeps gas prices low, which means that more manufacturing plants that rely on gas will be built in the United States

6) Is good for the environment because we will use less dirty energy sources such as coal and oil

7) Is a good partner for solar and wind energy because gas can be produced when the wind does not blow and the sun does not shine

8) Means that the U.S. can benefit from exporting natural gas to countries in Asia and Europe

9) Generates more income, and therefore more tax revenues to use for schools, hospitals, and other public services

10) Benefits individuals by paying them for their mineral rights

11) Strengthens the U.S. economy

#### Disadvantages

1) Delays the development of more sustainable and renewable sources of energy such as solar and wind energy

2) Uses up too much water, not leaving enough for the needs of the local area

3) Releases a gas (methane) that contributes to global warming

4) Uses chemicals that contribute to pollution of drinking water

5) Causes damages to human health and the environment

6) Results in more truck traffic

7) Results in wastes that are unmanageable

8) Reduces the quality of life in the communities located near the development

9) Contributes to earthquakes

| 10) | Causes toxic air pollution in communities near the development |
| --- | --- |
| 11) | Reduces the real estate values in the communities located near the development |

*2.2. An Overview of Scientific Assessments Regarding the Advantages and Disadvantages of Fracking*

With regard to scientific assessments, we read and compiled about 110 scientific, technical, and economic studies published from 2007 to 2020, including both original peer-reviewed papers and evidence-based reviews. None of the studies addresses all 22 claims covered in the paper but some addressed more than one claim. Actually, due to the specific characteristics of the claimed advantages and disadvantages, the sources of the literature are different. The advantages of fracking are mainly addressed in the economics-oriented literature, while the disadvantages of fracking are addressed mainly in the geological, epidemiological, and environmental literatures. We summarized the findings in the literature and classified each claimed advantage or disadvantage as either (a) supported by the weight of the available evidence, (b) not supported by the weight of the available evidence, or (c) there is inadequate evidence to assess the claim.

We then compared the public and scientific assessments of each claim for possible disconnections. We attempted to explain why respondents might regard some claimed disadvantages as highly important, even though the weight of the scientific evidence is inadequate to assess them or does not support them.

### 3. Results

Figure 1 presents the percentage of the respondents who perceived that a claimed advantage or disadvantage was an "extremely important" factor in their feeling about fracking. The advantages/disadvantages are ranked by the frequencies of the respondents.

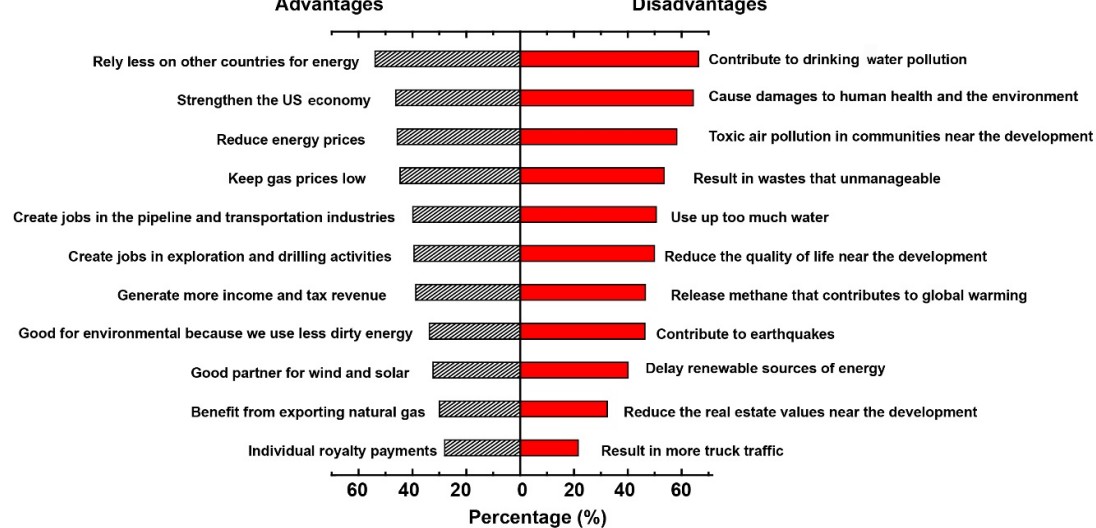

**Figure 1.** Percentage of Respondents who Classify Each Claimed Advantage and Disadvantage as "Extremely Important" by Respondents (n = 2833).

We find that the advantages and disadvantages are not equally important to respondents. Energy independence (the exact claimed advantage in the survey is "rely less on other countries for energy") and a stronger economy (the exact claimed advantage in the survey is "strengthen the US economy") are most frequently reported as extremely important (54% and 46%, respectively). Energy independence is the only claimed advantage perceived as "extremely important" by over half of the respondents. The other

nine advantages are seen as extremely important by substantial minorities of the respondents. Benefits paid to individuals for their mineral rights is least frequently cited as extremely important, by slightly over a quarter of respondents.

Compared to the perceptions of advantages, the evaluation of the claimed disadvantages has a generally greater spread. Five out of eleven disadvantages are perceived as "extremely important" by at least half of the respondents. Drinking water contamination is most likely to be reported as "extremely important," with 66% of respondents perceiving it as "extremely important". The next most-likely disadvantage is damage to human health and the environment, indicated as "extremely important" by 63% of respondents. Only 21% of respondents perceive more truck traffic as an "extremely important" disadvantage that shapes their feeling about fracking.

Another interesting finding is that respondents rank more of the disadvantages as "extremely important" than the advantages. Thus, as Figure 1 indicates, there is a wider spread among respondents with regard to disadvantages than advantages. We interpret this finding to mean that respondent reactions to the claimed advantages are more evenly bunched in perceived importance whereas the claimed disadvantages trigger more disparate reactions in terms of their importance. The differences between the highest and lowest ranked advantages and disadvantages are significant (p-value < 0.001).

There were no significant interstate variations in the ranking of public perceptions of the importance of the claimed advantages and disadvantages. The Spearman's correlation coefficients, which compare the rankings of the eleven advantages (or disadvantages) for any two states, range from 0.891 to 0.991 (and 0.882 to 0.982 for disadvantages). In each state, most respondents perceive energy independence as an "extremely important" advantage and drinking water contamination as an "extremely important" disadvantage of fracking. Yet we found significant differences across states when examining the pattern for each claimed advantage or disadvantage. For example, significantly fewer respondents residing in California perceived "rely less on other countries for energy" as extremely import in their support for fracking compared to the other five states. More respondents residing in Ohio and Texas perceived "creates jobs in the pipeline and transportation industries" as extremely important in their support for fracking compared to respondents from California and New York.

### 3.1. Advantages

### 3.1.1. Energy Independence

The term "energy independence" has been mostly and historically used as less dependence on imported oil [28]. We use the term beyond oil to include other energy/fuel sources, e.g., natural gas, and define the term narrowly as less imports of energy sources from foreign countries. The exact wording of the claimed advantage displayed in the survey is "Natural gas production, increasingly accomplished with fracking means that the United States can rely less on other countries for energy". Decreasing natural gas imports and becoming a net natural gas exporter improves energy independence. Imports of natural gas (primarily by pipeline via Canada) were, net of exports, steadily rising in the U.S. until fracking began to ramp up around 2007. Since then, imports have declined 37% in 6 years. The recession contributed to the decline but the decline has continued through the post-2009 recovery. The Energy Information Administration (EIA) projects that the U.S. will become a net exporter of liquid natural gas in 2016 and an overall net exporter of natural gas before 2020 [29]. Thus, the available evidence supports the claim that fracking helps the U.S. accomplish energy independence.

### 3.1.2. Strengthening the U.S. Economy

We discuss three potential effects of fracking on the U.S. economy. First, an income effect results from a lower gas price: the consumer saves on gas and can spend more on other goods. Consumers in the U.S. are saving $20 to $40 billion per year on natural gas

due to the lower gas prices induced by fracking [30]. Those savings come primarily in the form of lower prices for electricity and other goods and services that are derived from natural gas. Second, a substitution effect results from lower gas prices: cheap gas reduces the cost of producing other goods (e.g., chemicals, plastics, steel and so forth). Published reports have found positive substitution effects of fracking [31,32]. Nonetheless, because gas-intensive sectors (such as chemicals, primary metals, and the paper and print sectors) account for only a small share of manufacturing and the overall U.S. economy, the substitution effect may not be large on an economy-wide basis [33]. Third, a stimulus effect results from fracking activities: fracking will create jobs and increase the demand for inputs directly in gas production and indirectly in the manufacture of gas-intensive goods. While a positive stimulus effect is possible in 2007–2014, when the U.S. economy was operating below full employment, fracking may contribute little growth when the economy is fully recovered because resources in the economy will simply be shifted from other industries towards shale gas without producing a net growth in the economy [34]. Thus, in the long term, the stimulus effect of fracking is expected to diminish or disappear entirely [35]. The size of the near-term stimulus effect is not clear. Several studies found a negligible positive (0.46–0.48% of GDP) stimulus effect on the U.S. economy or a negligible to substantial positive effect on regional economies [36–40]. Most economic reports employ IMPLAN input-output models that do not account for the fact that a growth in shale gas may cause a reduction in the production of coal, nuclear, and renewables.

Overall, the available evidence supports the claim that fracking has helped the national economy, though more rigorous economic research and more precise estimates of the magnitude of the effects and the geographic location of the changes are needed.

### 3.1.3. Reducing Energy Prices

The growth of fracking has led to a decline in both natural gas and coal prices, the primary inputs to electricity production in the United States. Contrasted with a baseline without enhanced natural gas supply, projected electricity prices for the next 20–30 years are lower in the scenario with fracking [41,42].

Insofar as natural gas and oil are somewhat substitutable in many applications, more plentiful gas supplies (and lower gas prices) might cause some users to shift from oil to natural gas [43]. Under this scenario, one might expect the resulting decline in oil use in the U.S. to exert downward pressure on oil prices [44,45]. However, unlike natural gas (which is cumbersome and expensive to transport), oil is a global commodity that is relatively easy to transport. Thus, it is not obvious that growing gas supplies in the U.S. will lead to a meaningful drop in the global oil price. Hartley et al. [44] find evidence that U.S. natural gas prices tend to respond to movements in the international crude oil market, but the reverse is not observed.

The new technologies used for fracking are now increasingly used to boost oil production in the U.S., e.g., in North Dakota's Bakken [46] and Texas's Eagle Ford plays [47]. The boost in U.S. production, coupled with weaker demand from China, Europe, and the U.S., caused a sharp drop in the world price of oil in 2018 and then a huge drop in 2020 due to the coronavirus pandemic. It is not clear how long the lower oil price will last, as in its recent report, International Energy Agency [48] forecasts that Saudi Arabia and other OPEC countries might maintain the current supply in the next year. World oil prices ran up past $80 per barrel in May 2021 following OPEC decisions to moderate production and a surge in global demand after the easing of pandemic restrictions

Overall, the weight of the available evidence supports the claim that fracking is curbing the growth in electricity prices. There is inadequate evidence to assess fracking's impact on the world oil price, though the recent application of new technology to U.S. oil production does seem to be having a downward influence on the world oil price.

### 3.1.4. Keeping Gas Prices Low

Expanding U.S. production of natural gas has reduced U.S. prices of natural gas, and those prices are projected to stay much lower than prices in Europe and Asia for many years to come [30]. Figure 2 depicts the history of U.S. natural gas prices. After 2008, there is a sharp drop of natural gas wellhead price (from $8/Mcf to $3.8/Mcf in nominal dollars). However, it is inappropriate to attribute this lower price entirely to increasing supply because the Great Recession (2007–2009), which occurred during this period, resulted in a reduction of demand for gas. The U.S. economy has slowly but steadily recovered since mid-2009, with 2013 being the best year of the recovery [49]. Although natural gas prices have increased somewhat compared to their lowest levels of 2007–2008, they have not increased nearly as rapidly as overall GDP and remain much lower than prices in Europe and Asia [29,50].

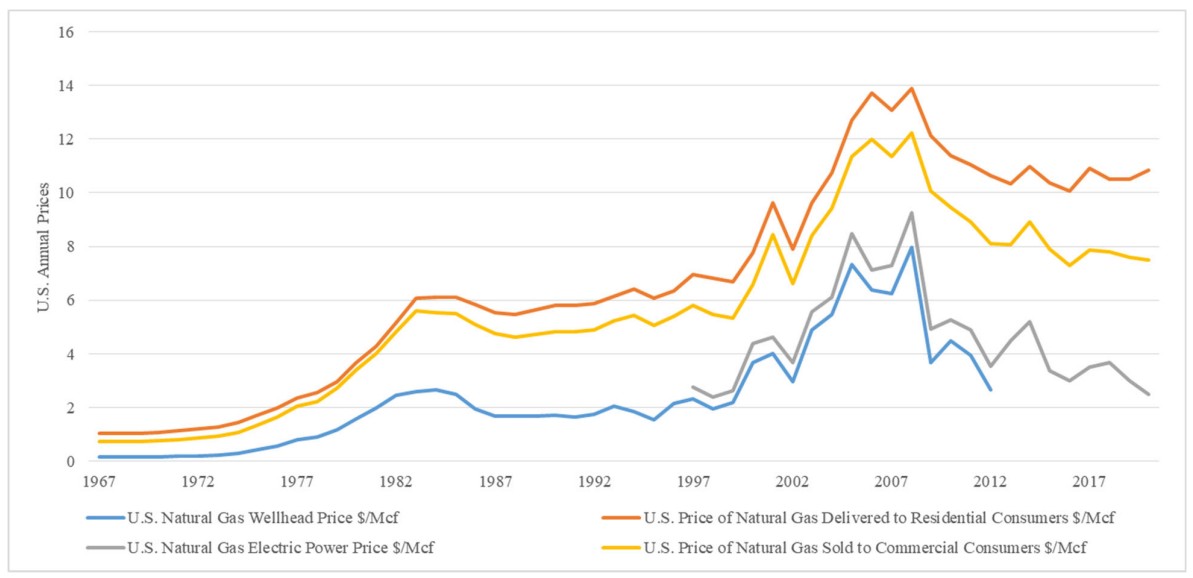

**Figure 2.** U.S. Natural Gas Prices, 1967–2020. Data Sources: U.S. Energy Information Administration 2021.

Overall, the weight of the evidence supports the claim of lower natural gas prices in the U.S. due to the expansion of fracking.

### 3.1.5. Creating Jobs in the Exploration and Drilling Activities

The workforces associated with exploration are very small; for the most part they are highly educated and specialized and will most likely come from outside of the development area. Due to the relatively short time period of this phase of fracking, workers will likely maintain a transient or temporary residency status in the development area.

The drilling phase is by far the most industrially active and labor intensive and will require the largest amount of both transient and local workforces. For drilling, both unskilled or semi-skilled and skilled positions are required. In addition, drilling activity does create "secondary" workforce demands in the local economy [39,51–53].

From 2007 to 2013, total annual employment in the U.S. decreased by 2.7% [54]. In contrast, the total mining, quarrying, and extraction employment in the oil and gas industry increased by 23.1% [54].

Overall, the weight of the evidence supports this claimed advantage of fracking.

### 3.1.6. Creating Jobs in the Pipeline and Transportation Industries

Expanding fracking creates jobs in supportive industries such as pipeline and transportation as excavators, truckers, pipeline installers, and material handlers. Based on the

Bureau of Labor Statistics (from 2007 to 2013) [54], the annual average employment in pipeline transportation and construction increased from 123,613 to 159,289 (i.e., 28.86%).

Several reports estimated large positive employment effects of fracking in impacted states, as reviewed by Kinnaman [35]. Peer reviewed studies using statistical approaches generally find negligible to moderate effects in local economies [38,53]. Overall, the weight of the evidence supports this claimed advantage of fracking. However, note that the input-output models used in the reports mentioned above do not always account for any displaced employment in the coal, nuclear, and renewable sectors (including their supply chains), neither do the statistical approaches.

### 3.1.7. Generating More Income and More Tax Revenues

Fracking may generate more income by a combination of greater demand for labor and an increase in the number of jobs or rent or royalty payments to private and public resource owners [55]. Although the long run effects of fracking on income and distribution of income are not clear [55,56], positive short-term effects on income levels have been reported in case studies [38,53,57].

Governments may have more tax revenue from severance taxes, local property taxes, sales taxes, and direct payments from the gas industry (i.e., fees, permits, and licenses). The existing literature finds a positive effect of fracking on government tax revenue [58,59]. Newspaper articles and research reports also found net positive impacts of fracking on income and government revenue [60,61]. One exception is Kinnaman [35], who found quite negligible impacts of shale gas extraction on tax revenue with a difference in difference approach.

Overall, the weight of the evidence supports a positive effect on income and government revenues. Short term impacts are better understood than long term impacts. Note that the available studies do not account for any lost income and tax revenue associated with declining production of coal, nuclear energy, and renewables and their supply chains.

### 3.1.8. Good for the Environment because of Less Use of Dirty Energy Sources

Since 2007, natural gas has emerged as a strong competitor to coal in the electric power sector. The percentage of electricity produced by coal in the U.S. has declined from about 48% in 2007 to 39% in 2013, with the share of gas-fired electricity rising by a similar amount, from 21% in 2007 to 27% in 2013. Forecasts suggest this competition will last at least until 2020 [29,50].

Both the coal mining and gas-development processes create environmental risks but the environmental footprint of coal mining (e.g., mountaintop and deep mining in West Virginia/Kentucky and strip mining in Wyoming) is generally considered to be more serious (e.g., more land is affected and disruptions are more extensive) than the footprint of fracking [62]. With respect to local air quality, gas-fired power plants emit almost no sulphur dioxide or mercury and fewer nitrogen oxides and particulate matter than coal-fired power plants [63].

However, cheaper gas has also hurt clean sources of electricity but to a far lesser extent than coal. Older nuclear plants are being retired and fewer new nuclear plants are under consideration. The growth of renewable sources of electricity may also have been attenuated, but less so in those states that mandate a minimum share of the electricity market for renewables. Nonetheless, the primary effect of cheap natural gas in near term electricity markets has been to induce a shift from coal to natural gas.

If one considers only coal replacement and impacts on local air quality and land impacts, there exists adequate evidence to confirm this advantage.

### 3.1.9. Good Partner for Solar and Wind Energy

Solar and wind are intermittent sources of electricity, which is a main hindrance that prevents the widespread diffusion of solar and wind energy. The development and diffusion of solar and wind energy is extremely challenging given the lack of cheap energy storage batteries in large scale or other storage technologies to cope with the grid supply and demand fluctuation. Using gas-electric energy as a backup source of power for wind and solar has several advantages. Gas-fired power plants have lower capital costs than nuclear or coal. The application of advanced production techniques that include hydraulic fracturing and horizontal drilling will make the continued use of abundant and low cost natural gas possible. Gas is also flexible in its deployment. Gas turbines can be turned on and off quickly to meet fluctuating power demands; in contrast, large boilers based on coal and nuclear power can take hours to turn on, resulting in large efficiency losses. In other words, it is more challenging to integrate coal or nuclear with intermittent sources of power than it is to integrate gas. In addition, the need for an abundant supply of natural gas-fired plants as good partners for solar and wind energy is also driven by the retirement of coal plants and accelerated retirement of nuclear plants. Because wind and solar combined with natural gas present a good combination of efficiency and greenness, the National Renewable Energy Laboratory [64] projects that such a combination will be the typical energy supply in 2050. Verdolin et al. [65] found a robust correlation between gas-fired power generation capacity and renewable capacity and highlighted the complementarity of investments in different generation technologies

The weight of evidence supports this claimed advantage, assuming no breakthrough in energy storage technology that would make renewables self-sufficient (without any need for gas as a partner).

### 3.1.10. Benefit from Exporting Natural Gas

Natural gas prices in Europe and Asia are much higher than they are in the U.S. [66]. Tsafos [66] projected that, though the natural gas prices across different regions are recently converging, European and Asian prices should revert back and being higher than those in the U.S., but may not revert back to where they were in the last decade. Europe is looking to reduce its dependence on Russian gas. Japan is eager for natural gas given the phase-out of the country's nuclear power plants. An opportunity for the U.S. lies in the export of natural gas to Europe and Asia. The United States has become a net exporter of natural gas since 2017; the total annual U.S. natural gas exports reached the highest on record in 2019 (4.66 Tcf), and the U.S. exported to about 38 countries [67]. The U.S. will gain net economic benefits from more exports of natural gas [68]. As long as the sale prices of natural gas are high enough to justify the costs of drilling, facilities, liquefactions, transportation, and other administrative costs, the U.S. can benefit economically from exporting natural gas. If natural gas exports increase rapidly, the price of natural gas in the U.S. may rise due to the diminished gas supplies in the U.S. market. The export of gas will increase economic activity, but the income gained from exports depends on how the rate of increase and level of export are determined and the price elasticity of the natural gas supply [69].

Overall, the weight of evidence supports the claim that there will be a net benefit from exporting natural gas from the U.S. to Europe and Japan.

### 3.1.11. Benefits Individuals by Paying Them for Their Mineral Rights

Mineral rights come with ownership of the property and are the rights to any minerals that lie underneath the surface. Property owners can legally sell these rights. Natural gas industries pay property owners an amount of money to buy or lease mineral rights when seeking land to drill on. Many factors will affect the amount of the leasing bonus and royalty payments, such as the production rate, natural gas price, and the royalty percentage. Expanding fracking led to an increase in average bonus payments, for instance,

from \$30/acre in 2005 to \$2400/acre in 2008 in the Marcellus region [70]; from \$200 to \$28,000 per acre in southern Texas; by 1.75 times in 2012 in West Virginia; they more than doubled in Pennsylvania; and \$8000 per acre was recorded in Ohio in the third quarter of 2012 [71].

Overall, the weight of evidence supports this claimed advantage.

### 3.2. Disadvantages

### 3.2.1. Drinking Water Contamination

Horizontal drilling and hydraulic fracturing are associated with multiple risks of drinking water contamination. We focus on the risks due to chemical additives entering potable water systems. Hydraulic stimulation uses large volumes of sand and water blended with a small amount of chemical additives to produce hydraulic fracturing fluid that is injected into the well. These chemical additives change the rheology of the fluid and may inhibit biological growth that can reduce the flow paths for the oil and gas [72].

Chemical additives can contaminate drinking water through four possible mechanisms. First, fracturing fluid might migrate into underground sources of drinking water if the integrity of a well is compromised during or after the fracking operations. However, the incidence rate of faulty casing in fracking is low, about 1 to 3% [73]. Second, migration of fluids from the productive reservoir could directly contaminate groundwater. However, studies with tracers have demonstrated that fracking fluids typically become trapped in the shale, migrating less than a few hundred feet from the horizontal portion of the well bore [74]. Third, surface leakage and spillage of fluids and solids from holding tanks, pits, and other containers can lead to surface and/or groundwater water contamination. This pathway is plausible without proper management. Fourth, the ("flow back") wastewater produced by natural gas production, which may contain drilling fluids, can pollute drinking water if the wastewater is not managed properly on the surface or in the subsurface.

Previous studies support the causal relationship between drinking water contamination and fracking [75,76], while others dispute this finding [77]; but previous studies focus on methane in groundwater, not the chemical additives in frac fluids. Jackson [78] has shown that the flow back water sometimes contains more naturally occurring methane, typically at higher concentrations than chemical additives, and thus may be of greater concern for drinking water contamination than the chemical additives from a relative risk point of view. The Royal Society (UK) concludes that the risks of chemical additives can be managed effectively as long as regulators and operators establish, implement, and enforce best practices [79]. For example, in Texas, 16,000 horizontal shale gas wells with multi-stage fracturing stimulations (1992–2008) did not find a single case of groundwater contamination from site preparation, drilling, well construction, completion, stimulation, or production operations [80]. In addition, a recent EPA assessment of fracking and drinking water contamination identified 151 out of 457 spills of chemical additives or fracturing fluids. Over half of these 151 spills were caused by equipment failure (mainly storage spillover) and human error. Though specific instances of chemical additives and water pollution were found, the EPA did not find evidence of widespread and systematic impacts of fracking on drinking water in the U.S. [81].

There are potential pathways for the chemical additives to contaminate drinking water, but the weight of current evidence is inadequate to support this claimed disadvantage. Rigorous studies monitoring water quality at multiple sites before and after fracking are necessary to gauge this claimed disadvantage.

### 3.2.2. Damage to Human Health and Environment

The framing of this claimed disadvantage is too broad to relate it to a single specific body of literature. This section will evaluate the key evidence of human health and environmental impacts.

Previous literature has discussed the potential health risks associated with fracking [82–84], but not identified the set of possible health outcomes, let alone the magnitude and distribution of the risks [85]. Current health studies tend to be constrained by a specific spatial and temporal scope of available monitoring data or rely on traditional emission inventories and atmospheric dispersion and chemistry models to determine the magnitude of human exposures and related health risks [86,87]. Prior research has not demonstrated a causal relationship between fracking and specific health effects [88,89]; more often data are drawn from anecdotal reports of adverse health outcomes among industrial workers or residents living nearby [90].

A recent report from the International Risk Governance Council itemized potential environmental risks to the soil, water, air, and ecosystems [91]. First, land impact from fracking (mainly from roads, pipeline right-of-ways, production, and gathering facilities) is larger for fracking than for conventional natural gas development. Direct land use impact may induce a secondary impact on land quality, habitat loss, ecosystem fragmentation, and inadequate surface rehabilitation [85]. Second, fracking creates dust and diminishes air quality (temporarily) from the construction of the well pad, transportation, drilling, hydraulic fracturing, and use of heavy machinery (see 3.2.3). Third, there are plausible pathways through which fracking might contaminate water (see 3.2.1). A good suite of studies exists on the impact of fracking on surface water impacts, though inadequate evidence exists about the potential adverse effects of fracking on groundwater [74,83,92].

To sum up, there is inadequate evidence about damages to public health and the environment. Firm conclusions cannot be drawn until researchers complete more rigorous empirical studies [85,93].

### 3.2.3. Causing Air Pollution in Nearby Communities

The fracking process releases methane, the primary constituent of natural gas, but also alkanes, benzene, and other aromatic hydrocarbons. Recent studies found increasing concentrations of methane and non-methane hydrocarbons and air toxics near drilling sites [94–96]. A recent study found the highest concentrations of over 20 potentially toxic hydrocarbons downwind less than 500 feet from well pads during flow back operations [82].

Overall, the weight of evidence supports this claimed disadvantage. Further study is needed to explore the contribution of fracking to the concentration of individual chemicals, complex mixtures of chemicals, and secondary air pollutants locally and downwind. A special focus on persistent, bioaccumulative, toxic chemicals (PBTs) is justified because they can be transported long distances.

### 3.2.4. Resulting in Unmanageable Wastes

Fracking generates large amounts of wastewater that contain potentially hazardous pollutants such as salts, organic hydrocarbons, additives, and naturally occurring radioactive materials.

Wastewater treatment and disposal options include industrial treatment, disposal at publicly owned treatment works (POTW), reuse/recycle, and injection and disposal of wastes into deep underground wells [81,97,98]. Over 98% of produced water is injected underground [99], and most fluids are intended to remain trapped in the shale, a long-term form of waste management. The effectiveness of this method has been demonstrated in multiple studies supported by the Department of Energy. Implementing effective regulations for injecting produced water has significantly improved wastewater management in Texas [99].

Another study of the Marcellus Shale found that fracking generates less wastewater than before and, thus, if industry modifies its waste management practices, potential risks can be addressed, especially when regulatory or economic drivers are in place [100]. To encourage use of best practices that are beyond the mandatory minimum, voluntary

standards and certification systems are recommended or are under development [101,102].

The weight of the current evidence suggests that wastes are in fact manageable, if best management practices are followed by the industry.

### 3.2.5. Large Amount of Water Use

Hydraulic fracturing requires a large amount of water injection along with sands and chemicals [103]. The Department of Energy [104] states that 10,000–20,000 m$^3$ of water per well are required for shale formations. The water is typically taken from local sources due to the high cost of water transport. The consumption of freshwater for fracking could lead to water supply diminution. Regions where freshwater supplies are constrained are of particular concern. Since 2011, there has been a significant increase in recycling of wastewater, but widespread recycling of wastewater remains to be accomplished.

The weight of evidence supports this claimed disadvantage.

### 3.2.6. Reducing the Quality of Life in the Nearby Communities

Quality-of-life concerns include increasing truck traffic, greater noise levels from drilling operations, industrial odors, and poor air quality. A rapid population influx could also put pressure on pre-existing resources and community infrastructure. Quality-of-life impacts are related to residential proximity to fracking sites. The current literature, based on interviews with key stakeholders, supports the negative effects of shale gas development on the quality of life in boomtowns [105,106].

The weight of the evidence supports this claimed disadvantage.

### 3.2.7. Methane and Global Warming

Fracking is applied in natural hydrocarbon seeps. The seeps leak methane when they leak oil and gas naturally [107]. In addition, flow back water is accompanied by large quantities of methane (so called "fugitive methane"), which is far more than could be dissolved in the flow back fluids. Lastly, methane is emitted when the plugs set to separate fracturing stages are drilled out.

Recent studies do not agree on the magnitude of methane emissions from fracking. Howarth et al. [108] estimated 3.6% to 7.9% of the methane from shale-gas production escapes to the atmosphere in venting and leaks over the lifetime of a well. They found further that the methane emissions from fracking are more than those from conventional gas on any time horizon and more than those from coal on the 20-year horizon, but are comparable to coal when compared over 100 years. Cathles et al. [109] conclude oppositely that shale gas has a GHG footprint that is half to a third that of coal. McJeon et al. [110] shows an increase in climate forcing associated with the increased use of abundant gas. More reliable data are needed to improve the accuracy of emission inventories and identify high-emitted sources [111].

The EPA and the states have recommended technologies and practices that natural gas producers can use to minimize fugitive methane, but only in some states are the measures mandatory for existing operations. The EPA has required methane controls at new gas production sites and is working with the industry on voluntary regulations for existing sites.

The weight of the evidence supports this claimed disadvantage, although future methane regulations may minimize the importance of this disadvantage.

### 3.2.8. Induced Seismicity

The injection of wastewater can cause a pore pressure increase, which, when transmitted into the seismogenic fault zone, lowers the fractural stress and reduces the forces that keep a fault locked. The result can be an earthquake. Note that the wastewater itself

is not necessarily reaching the fault. Many studies have compared the timing of an earthquake, the disposal well location, and injected wastewater volume to help establish a positive correlation or a causal inference (e.g., Justinic et al. [112]; Rubinstein et al. [113]; Keranen et al. [114]).

Frohlich et al. [115] also found that increasing fluid extractions are directly associated with small-magnitude seismic events in Eagle Ford Shale. Recently, several state regulatory actions in Ohio and Oklahoma, along with ongoing legal challenges to the practices of high-volume wastewater injection, presume the plausibility of this disadvantage.

Overall, the current scientific evidence supports the claim that deep-well injection of wastes and, to a lesser extent, extraction of gas is associated with measurable seismic events. Most of the events have not been associated with significant damages.

### 3.2.9. Delaying the Solar and Wind Development

The expansion of fracking tends to depress natural gas prices, which in turn diverts some demand away from wind and solar [116,117]. While shale gas may reduce coal use in the short run, it may slow the process of decarbonization, primarily by delaying renewable energy development in the long run. A variety of factors could mitigate this perverse effect: a liberal policy toward natural gas exports, which would tend to boost natural gas prices in the U.S.; a requirement that a carbon capture and storage (CCS) rule applied to gas-fired power plants (a more stringent stance than the EPA has proposed); and a continuing trend toward renewable energy requirements at the state level.

Overall, the current evidence lends credence to this claimed disadvantage.

### 3.2.10. Reducing Real Estate Values in the Nearby Communities

The impacts of fracking on property values vary with geographic scale, water sources, well productivity, visibility, royalties, and other factors [118,119]. On the one hand, the perceived risks of fracking related to groundwater contamination can be expected to negatively impact real estate values, even though the value of homes with piped water may increase due to the royalty payments and lease payments [118]. The adverse quality-of-life impacts of fracking may also depress property values. On the other hand, if neighbors benefit from fracking by royalty payments, then property values may rise as energy service providers seek contracts to develop more land in the neighborhood. Moreover, the increasing in-migration, employment, and economic activity associated with fracking may boost property values, but such effects may diminish with time [118].

There is inadequate evidence about this disadvantage to draw any conclusion.

### 3.2.11. Resulting in More Truck Traffic

Truck traffic near drilling sites can increase for several reasons. First, large amounts of water, sand, chemicals, and drilling equipment/materials need to be trucked to the site. A Marcellus Shale well requires about 1000 truckloads of materials to drill and complete [40]. Second, the wastewater produced by fracking must also be trucked off-site, unless a pipeline or rail line is available. Though the industry is making an effort to minimize the reliance on truck traffic, trucks are still the dominant transportation media. Increasing truck traffic may damage secondary roads, increase risks of truck accidents, and lower quality of life nearby due to more dust and noise.

The weight of the evidence supports this claimed disadvantage of fracking.

## 4. Comparison of Public and Scientific Assessments

Last, we compared public evaluation of each claimed advantage and disadvantage to current scientific evaluation. We gave emphasis to the claims that respondents frequently cited as "extremely important" to their feeling about fracking.

Table 3 compares the perspectives of the on-line survey respondents to our classifications of the weight of scientific evidence for each claimed advantage and disadvantage.

We acknowledge that there are limitations associated with a three-way classification of complex evidence patterns, but we found the classification useful as a straightforward and objective way to look for potential disconnections between public and scientific assessments.

**Table 3.** Classification of Eleven Advantages and Eleven Disadvantages of Expanding Natural Gas Production Using Fracking.

| Rank | Advantages | Percentage of Respondents Perceive the Claim as "Extremely Important" | The Weight of the Evidence Lends Credence to This Claimed Advantage/ Disadvantage | There is Inadequate Evidence about This Claim to Draw Any Conclusion | The Weight of the Evidence Does Not Lend Credence to This Claimed Advantage/ Disadvantage |
|---|---|---|---|---|---|
| 1 | Means that the U.S. can rely less on other countries for energy | 53.86% | √ | | |
| 2 | Strengthens the U.S. economy | 46.15% | √ | | |
| 3 | Reduces energy prices | 45.60% | √ | | |
| 4 | Keeps gas prices low | 44.56% | √ | | |
| 5 | Creates jobs in the pipeline and transportation industries | 39.86% | √ | | |
| 6 | Creates jobs in exploration and drilling activities | 39.40% | √ | | |
| 7 | Generates more income, and therefore more tax revenues | 38.71% | √ | | |
| 8 | Is good for the environment because we use less dirty energy sources. | 33.71% | | √ | |
| 9 | Is a good partner for solar and wind energy | 32.32% | √ | | |
| 10 | Means that the U.S. can benefit from exporting natural gas | 29.96% | √ | | |
| 11 | Benefits individuals by paying them for their mineral rights | 28.00% | √ | | |
| | **Disadvantages** | | | | |
| 1 | Uses chemicals that contribute to pollution of drinking water | 66.31% | | √ | |
| 2 | Causes damages to human health and the environment | 64.33% | | √ | |
| 3 | Causes toxic air pollution in communities near the development | 58.20% | √ | | |
| 4 | Results in wastes that is unmanageable | 53.56% | | | √ |
| 5 | Uses up too much water, not leaving enough for the needs of the local area | 50.58% | √ | | |

| | | | | |
|---|---|---|---|---|
| 6 | Reduces the quality of life in the communities located near the development | 49.92% | √ | |
| 7 | Releases a gas (methane) that contributes to global warming | 46.54% | √ | |
| 8 | Contributes to earthquakes | 46.37% | √ | |
| 9 | Delays the development of more sustainable and renewable sources of energy such as solar and wind energy | 40.07% | √ | |
| 10 | Reduces the real estate values in the communities located near the development | 32.38% | | √ |
| 11 | Results in more truck traffic | 21.58% | √ | |

Overall, public opinion about the advantages and disadvantages of fracking are largely consistent with the weight of scientific evidence (ten out of eleven in the case of advantages; seven out of eleven in the case of disadvantages). Contrary to prior expectations that benefit/risk as feelings would differ from benefit/risk as analysis, the current scientific evidence lends credence to the three claimed advantages of fracking that are most frequently considered to be "extremely important" to respondents.

Evidence of disconnection between the public and scientific assessments primarily lies in the case of the disadvantages. There is inadequate evidence to assess the claim that fracking causes drinking water contamination from chemicals or damage to human health and the environment—the two claimed disadvantages of fracking that are most frequently considered "extremely important". Further, the claim that fracking results in unmanageable wastes, which was evaluated as "extremely important" by over half of the respondents, is not supported where best practices are followed.

## 5. Conclusions and Discussion

The most interesting potential disconnections between public perceptions and the evidence-based classifications concern the two disadvantages most frequently cited as extremely important: drinking water pollution from chemicals and damages to human health and the environment. We offer possible explanations for the disconnection based on risk perception theory, language interpretation in the survey instrument, and possible measurement error.

Slovic [6] showed that technical experts and lay people have different foci when they rate the risks of hazardous activities and technologies. Technical experts focus on the plausibility of scientific theories and evidence-based numeric estimates of frequency and severity of harm. In contrast, lay people relate risk perceptions to qualitative characteristics of hazards/technologies, namely "dread risk" and "unknown risk", and their perceived acceptability. The top two disadvantages of fracking are readily understood in terms of dread and the unknown. Chemical pollution of drinking water is a classic source of dread because clean water is important for the survival of human beings. Because unknown risks tend to heighten the concerns among laypeople, the inadequate science about the top two disadvantages of fracking might exacerbate rather than dampen the importance ratings of lay respondents. Damages to human health and environment can also be readily understood in terms of dread and the unknown. Future research should examine whether risk-perception theory can provide a satisfactory empirical explanation for how the public reacts to the various risks of fracking.

Alternatively, the respondents may have reacted to the particular ways in which the top two disadvantages were framed in the questionnaire. On drinking water pollution, respondents may have reacted to pollution more generally rather than reacting to the

presence of chemical additives in water. As a result, they may have interpreted this disadvantage as "fracking may cause drinking water pollution" rather than "fracking may cause drinking water pollution due to its chemical additives." In other words, respondents may be voicing concern about drinking water contamination, regardless of whether chemical additives are the source of the problem. Future research should explore whether alternative framings of the drinking water-quality disadvantage yield similar or different ratings of importance.

It is difficult for the respondent not to rate "damages to human health and environment" as extremely important because the damages are taken as given and are not related in any specific way to the process of fracking. Public health and environmental protection are both widely-appreciated constructs. Future research might shed more light on the disconnection hypothesis by relating health and environmental concerns to specific aspects of fracking. Nonetheless, we emphasize that we found a number of cases in the review of media stories where health and environmental damages were generally associated with fracking, without any specific mention of a source, mechanism, or pathway.

Among the limitations of this study, we would like to highlight some difficulties surrounding the framing of the specific claimed advantages and disadvantages. Our intention was to formulate the claims as closely as possible to the accounts of advantages and disadvantages portrayed in the media and other sources that laypeople and opinion leaders might consult. In some cases, the framing was so broad or diffuse that it was difficult to relate it to a specific body of scientific literature (e.g., "damages to human health and the environment"). In other cases, a specific claim seemed to be a subset of a broader claim, which means the various claims overlap to some extent. Those overlaps caused some difficulties in organizing distinct reviews of the scientific evidence. Secondly, future research should target people living near drilling sites because they may have different perceptions than others. Thirdly, there is a degree of subjectivity in the weight-of-the-evidence interpretations. It would be interesting to determine whether another qualified team of experts would have classified the evidence the same way or differently than we did. The fact that we are trained in the subfields of environmental studies, geology, risk analysis, and political science may have contributed to our interpretation of the body of studies. Finally, both the public and the scientific assessments can change over time in response to real world events and new evidence. Our findings should be revisited over time.

At this early stage of the public debate about fracking, it is encouraging that the claims that are important to laypeople are generally aligned with the weight of the scientific evidence. We recommend more studies of public reaction to energy technology at an early stage of technology development.

Further research on how public education of the science and economics of fracking might exacerbate or attenuate perceptions of the advantages and disadvantages and further the public debate of fracking are needed.

**Author Contributions:** Conceptualization, Y.Z. and J.D.G.; methodology, Y.Z. and J.D.G.; data collection and validation, Y.Z.; formal analysis, Y.Z.; writing—original draft preparation, Y.Z.; writing—review and editing, Y.Z., J.A.R., and J.D.G.; visualization, Y.Z.; project administration, Y.Z. All authors have read and agreed to the published version of the manuscript.

**Funding:** This research received no external funding.

**Institutional Review Board Statement:** The study was conducted according to the guidelines of the Declaration of Helsinki, and approved (exempted) by the Institutional Review Board of Indiana University Human Research Protection Program (protocol code 1402757667 and date of approval 3/28/2014).

**Informed Consent Statement:** Informed consent was obtained from all subjects involved in the study.

**Data Availability Statement:** The survey data presented in this study are available on request from the corresponding author.

**Conflicts of Interest:** The authors declare no conflict of interest.

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
