# Peer review of "Contrasting Public and Scientific Assessments of Fracking"

_sustainability, doi:10.3390/su13126650_

Round 1
Reviewer 1 Report
Overall, I found that it is a good article, descriptive and analytical at the same time. It is illustrative of an important issue such as Fracking. The objective of contrasting Public and Scientific Assessments of Fracking is quite interesting and original.
Material and research methods are well set up. Perhaps, the survey on 'Public Perceptions of the Advantages and disadvantages of Fracking' conducted in 2014, it is a bit 'old', but I believe that still is relevant. Moreover, this is a minor point that could be complemented with a further survey in 2024, and compere results in ten years - 2014 and 2024.
There are good academic references, but perhaps a few more updates would be convenient to add.
To summarise, the message is informative and academic at the same time. Well written, and coherence, most importantly, it sends a honest message pointing out the limitations of the study.
Reviewer 2 Report
The content of the paper is interesting, and it exhibits valuable information about people’s attitudes to fracturing. I recommend publishing the paper once the author can address the following concerns.
1. Does the attitude of residents to fracturing vary in different states?
2. I assume the term “fracturing” is only related to fracturing in a gas reservoir, but the term “Energy independence” is mostly about petroleum or oil. Please specify the term in 3.1.1.
3. On line 253 “Saudi Arabia has not responded with a decline in their production, despite pressures from other OPEC countries to boost the price”. This opinion needs to be updated based on the new decision of OPEC.
4. On line 266 “from $8 to $3.8” please add volume units.
5. I am confused about chapter 3.1.9. “Good partner for solar and wind energy”. Theoretically, developing renewable energy while keeping the capacity of natural gas generators is economically infeasible. It is hard to say fracturing can benefit this opinion.
Reviewer 3 Report
This work presents well-written analysis that showed some differences between public and scientific assessment of hydraulic fracturing. I would recommend publishing this paper after several "minor revisions".
Line 129: Could authors add a reference here? Hydraulic fracturing (HF) is widely used not only stimulating gas shale formations and also to stimulate tight oil deposits.
Line 153: Could the authors please specify how exactly you claimed the weight of "available evidence"?
Line 141 Could authors please clarify why derived advantages and disadvantages are significant and valuable? Are they mostly widely spread concerns or authors used their methodology?
Line 198. Energy independence is referred to all energy sources not only to gas production.
Line 321 Same comment as on line 153. Could authors clarify how they defined "weights of the evidence"?
Line 362. Please add a reference.
Line 397: Could you please clarify your statement? The main goal of hydraulic fracturing is to stimulate the near wellbore region by inducing fractures and making it more permeable for oil and gas. How fracturing fluid can clog the pathways for gas?
Author Response
Response to Reviewer 3 Comments
The authors thank the reviewer for their valuable time and helpful comments on our manuscript. We really appreciate that the reviewer put a great deal of time and effort into helping us improve the manuscript. Below, we provide the detailed responses point by point in red.
Point 1: Line 129: Could authors add a reference here? Hydraulic fracturing (HF) is widely used not only stimulating gas shale formations and also to stimulate tight oil deposits.
Response 1: We listed two examples where hydraulic fracturing is used for shale oil production in section 3.1.3 with references. We also added a brief introduction of HF that is widely used in both gas and tight oil deposit with a reference on page 1.
Point 2: Line 153: Could the authors please specify how exactly you claimed the weight of "available evidence"?
Response 2: We searched academic research findings from a variety journals and research centers via google scholar for each claimed advantages and disadvantages.
- When there are relevant research and all the findings from various research are consistently supporting the claimed advantages or disadvantages, we clarified it as “supported by the weight of the available evidence”.
- When there are relevant research and all the findings from various research are consistently opposing to the claimed advantages or disadvantages, we clarified it as “not supported by the weight of the available evidence”.
- When there are no relevant research or there are relevant research yet the findings from various research are not consistent or there are a mixed findings, we clarified it as “inadequate evidence to assess the claim”.
- When there are relevant research and findings from the research support or do not support the claims from one perspective (e.g. short term vs. long term) and research on the other perspective is not clear, we clarified it as “supported by the weight of the available evidence” or “not supported” and add a caveat to highlight the ambiguous scientific results from the other perspective (e.g. section 3.1.7 generating more income and more tax revenue).
Point 3: Line 141: Could authors please clarify why derived advantages and disadvantages are significant and valuable? Are they mostly widely spread concerns or authors used their methodology?
Response 3: As written in the paper, we derived the claimed advantages and disadvantages from local media stories, from advocacy claims made by pro- or anti-fracking groups, and from think tank pieces. The derived advantages and disadvantages represented the mostly widely spread concerns or topics related to fracking by the public. In this way, we could comprehensively analyze the perceptions of fracking from various angles.
Point 4: Line 198. Energy independence is referred to all energy sources not only to gas production.
Response 4: The authors specified energy independence in section 3.1.1 and specifically explained the how fracking will reduce dependence on gas imports and production.
Point 5: Line 321 Same comment as on line 153. Could authors clarify how they defined "weights of the evidence"?
Response 5: Please see the author’s response to Point 2 comment for Line 153 above.
Point 6: Line 362. Please add a reference.
Response 6: We added a reference for this line and made an updated explanation.
Point 7: Line 397: Could you please clarify your statement? The main goal of hydraulic fracturing is to stimulate the near wellbore region by inducing fractures and making it more permeable for oil and gas. How fracturing fluid can clog the pathways for gas?
Response 7: It is not the fracturing fluid that clog the pathways for gas. The chemicals added to the fluid will help improve the permeability and inhabit the biological growth, e.g. fungus growth that can reduce the flow paths for the oil and gas. To avoid such confusion, we rewrote the sentence in the manuscript as: “These chemical additives change the rheology of the fluid and may inhibit biological growth that can reduce the flow paths for the oil and gas”.
In addition, chemical additives are added to make the frack fluid behave in desired manner after it has physically fractured the formation rock surrounding the well bore and improved the permeability of the producing formation. If not designed correctly, it can however damage the formation and reduce the permeability. This can happen by introducing microbial organisms that can flourish and reduce the flow paths for the oil and gas or, it can chemically interact with be the formation fluid and rock chemistry in an adverse manner, again reducing the fracture flow created by the physical fracturing. Lastly, the fluid itself must be engineered to effectively flow back out of the artificial fractures (the “flow back” process) to make room for and promote the flow of oil and gas from farther out in the formation, through the new fractures, into the well bore.
